# The Medkit-Learn(ing) Environment: Medical Decision Modelling through Simulation

**Alex J. Chan**
University of Cambridge
Cambridge, UK
alexjchan@maths.cam.ac.uk

**Ioana Bica**
University of Oxford
Oxford, UK
ioana.bica@eng.ox.ac.uk

**Alihan Hüyük**
University of Cambridge
Cambridge, UK
ah2075@cam.ac.uk

**Daniel Jarett**
University of Cambridge
Cambridge, UK
daniel.jarrett@maths.cam.ac.uk

**Mihaela van der Schaar**
University of Cambridge
Cambridge, UK
mv472@cam.ac.uk

## Abstract

Understanding decision-making in clinical environments is of paramount importance if we are to bring the strengths of machine learning to ultimately improve patient outcomes. Several factors including the availability of public data, the intrinsically offline nature of the problem, and the complexity of human decision making, has meant that the mainstream development of algorithms is often geared towards optimal performance in tasks that do not necessarily translate well into the medical regime; often overlooking more niche issues commonly associated with the area. We therefore present a new benchmarking suite designed specifically for medical sequential decision making: the Medkit-Learn(ing) Environment, a publicly available Python package providing simple and easy access to high-fidelity synthetic medical data. While providing a standardised way to compare algorithms in a realistic medical setting we employ a generating process that disentangles the policy and environment dynamics to allow for a range of customisations, thus enabling systematic evaluation of algorithms' robustness against specific challenges prevalent in healthcare.

## 1 Introduction

Modelling human decision-making behaviour from observed data is a principal challenge in understanding, explaining, and ultimately improving existing behaviour. This is the business of decision modelling, which includes such diverse subfields as reward learning [1, 55, 35, 29], preference elicitation [32], goal inference [54], interpretable policy learning [28], and policy explanation [10]. Decision modelling is especially important in medical environments, where learning interpretable representations of existing behaviour is the first crucial step towards a more transparent account of clinical practice.

For research and development in clinical decision modelling, it is important that such techniques be validated robustly—that is, operating in different medical domains, guided by different environment dynamics, and controlled by different behavioural policies. This is difficult due to three reasons. First, the very nature of healthcare data science is that any learning and testing must be carried out entirely offline, using batch medical datasets that are often limited in size, variety, and accessibility [25, 39]. Second, directly using methods for time-series synthetic data generation is inadequate, as they simply learn sequential generative models to replicate existing data, making no distinction between

Submitted to the 35th Conference on Neural Information Processing Systems (NeurIPS 2021) Track on Datasets and Benchmarks. Do not distribute.

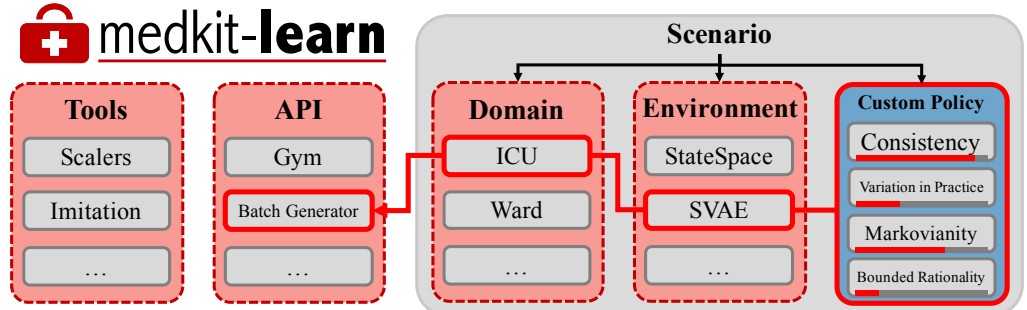

Figure 1: *Overview of Medkit.* The central object in Medkit is the *scenario*, made up of a *domain*, *environment*, and *policy* which fully defines the synthetic setting. By disentangling the environment and policy dynamics, Medkit enables us to simulate decision making behaviours with various degrees of Markovianity, individual consistency, bounded rationality and variation in practice. An example scenario is highlighted: ICU patient trajectories with environment dynamics modelled by a sequential VAE and a customised clinical policy. The output from Medkit will be a batch dataset that can be used for training and evaluating methods for modelling human decision-making.

environment and policy dynamics [20, 53, 14]. Because the environment and policy dynamics are entangled, such models do not allow for customisation of the decision making policy and thus cannot be used for evaluating methods for understanding human decision-making. Third, while various hand-crafted medical simulators have been proposed as stylised proofs-of-concept for research [43, 24, 16, 22], they often make unrealistic assumptions and simplifications that are unlikely to transfer well to any more complicated real-world setting. Moreover, these simulators do not directly allow obtaining offline data from different types of policy parameterisations.

**Desiderata:** It is clear that what is desired, therefore, is a tool that supports: (1) a variety of realistic environment models—learned from actual data, to reflect real medical settings), thus allowing simulation of (2) a variety of expressive and customisable policy models that represent complex human decision-behaviours; as well as (3) ensuring that the environment and policy components are disentangled—hence independently controllable.

**Contributions:** We present the Medkit-Learn(ing) Environment ("Medkit"), a toolbox and benchmarking suite designed precisely for machine learning research in decision modelling. Fulfilling all of the above key criteria, Medkit seeks to enable advances in decision modelling to be validated more easily and robustly—by enabling users to obtain batch datasets with known ground-truth policy parameterisations that simulate decision making behaviours with various degrees of Markovianity, bounded rationality, confounding, individual consistency and variation in practice. Moreover, to facilitate efficient progress in this area of understanding human decision-making, we have built Medkit to be freely accessible and transparent in data simulation and processing to enable reproducibility and fair benchmarking.

## 2 The Medkit-Learn(ing) Environment

Figure 1 gives an overview of the structure of Medkit, demonstrating a modular design philosophy to enable an ever-growing offering of scenarios as new algorithms and data become available. Medkit is publicly available on GitHub: `https://github.com/XanderJC/medkit-learn`. Written in Python and built using PyTorch [46] for a unified model framework, Medkit takes advantage of the OpenAI gym [9] interface for live interaction but has otherwise minimal dependencies.

### 2.1 Simulating Medical Datasets for Modelling Sequential Decision-Making

Our aim is to build generative models for the decision making process, that allow for full customisation of: (1) the environment dynamics, that model how the patient's state changes; and (2) the policy dynamics through which users can specify complex decision making behaviours.

Formally, we define a *scenario* as a tuple $(\Omega, \mathcal{E}, \pi)$, which represents the central component of Medkit that fully defines a generative distribution over synthetic data. A scenario comprises a medical

64 *domain*, $\Omega$ (e.g. the ICU); an *environment* dynamics model for sequential observations, $\mathcal{E}$ (e.g. a
65 linear state space model); and a *policy* mapping the observations to actions, $\pi$ (e.g. a decision tree).

66 Let $\vec{x}_T = x_s \cup \{x_t\}_{t=1}^T$ be the individual patient trajectories and let $\vec{y}_T = \{y_t\}_{t=1}^T$ be the clinical
67 interventions (actions). Here $x_s \in \mathcal{X}_s$ is a multi-dimensional vector of *static* features of the patient, e.g.
68 height, various comorbidities, or blood type - while $x_t \in \mathcal{X}$ a multi-dimensional vector representing
69 *temporal* clinical information such as biomarkers, test results, and acute events. Additionally $y_t \in \mathcal{Y}$
70 is a further possibly multi-dimensional vector representing the actions and interventions of medical
71 professionals, for example indicators for ordering tests and prescribing treatments.

72 We propose modelling the joint distribution of the patient features and clinical interventions $p(\vec{x}_T, \vec{y}_T)$
73 using the following factorisation:

$$p(\vec{x}_T, \vec{y}_T) = \underbrace{P_{\mathcal{E}}^{\Omega}(x_s) P_{\mathcal{E}}^{\Omega}(x_1|x_s) \prod_{t=2}^{T} P_{\mathcal{E}}^{\Omega}(x_t|f_{\mathcal{E}}(\vec{x}_{t-1}, \vec{y}_{t-1}))}_{\text{Environment}}$$
$$\times \underbrace{Q_{\pi}^{\Omega}(y_1|x_s, x_1) \prod_{t=2}^{T} Q_{\pi}^{\Omega}(y_t|g_{\pi}(\vec{x}_t, \vec{y}_{t-1}))}_{\text{Policy}}, \tag{1}$$

74 where the distributions $P_{\mathcal{E}}^{\Omega}(\cdot)$ specify the transition dynamics for domain $\Omega$ and environment $\mathcal{E}$ and
75 $Q_{\pi}^{\Omega}$ represents the policy for making clinical interventions in domain $\Omega$, thus defining the decision
76 making behaviour. Note that the patient trajectories and interventions depend on the entire history
77 of the patient. The functions $f$ and $g$ are modelled to be distinct such that the focus on the past
78 represented in the conditional distributions may be different for both the policy and the environment.
79 While this factorisation allows for enough flexibility, we will often make use of a representation
80 given in the graphical model of Figure 2 which includes a hidden state of the environment $z_t$ to be
81 the underlying driver of the data. Note this is even more general and we recover equation (1) if $z_t$ is
82 simply set as $x_t$.

83 With the factorisation proposed in Equation 1 we
84 notice a clear separation between the *environment*
85 and *policy* dynamics so that they can be modelled
86 and learnt separately, with the *domain* defining the
87 "meta-data" such as the spaces $\mathcal{X}_s, \mathcal{X}$, and $\mathcal{Y}$. This
88 disentanglement between the environment and policy
89 components is not possible in current synthetic data
90 generation methods (as we explore in section 3). A
91 corollary to this makes for a useful feature of Medkit
92 - that we can then mix and match elements of the
93 tuple to create a variety of different scenarios that can
94 be extended easily in the future when new models
95 or data become available. This not only satisfies our
96 desiderata, but also enables Medkit users to generate
97 a variety of batch datasets with customisable policy
98 parameterisations (e.g in terms of Markovianity, re-
99 ward, variation in practice) and thus evaluate a range of methods for understanding decision-making.

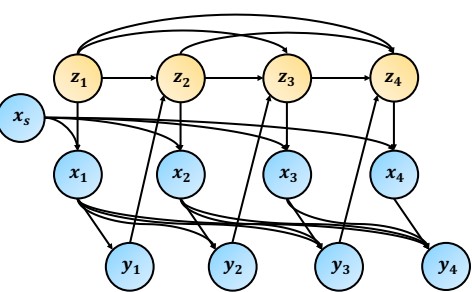

Figure 2: *Graphical model of the generative process we consider.* Usually there will be some hidden state of the environments that drives the actions and observations seen.

## 2.2 User workflow

100

101 Medkit was build to facilitate the development of machine learning methods for clinical decision
102 modelling. Medkit offers users the flexibility to obtain batch datasets $\mathcal{D}_{syn,\mathcal{E}}^{\omega}$ for any desired type
103 of parameterisation $\theta$ (e.g. temperature, Markovianity, reward) of the decision making policy $Q_{\pi_\theta}^{\mathcal{E}}$
104 and thus evaluate a wide range of methods for modelling sequential decision making. This includes
105 methods for recovering expert's reward function [11, 7], subjective dynamics [28] or interpretable
106 policies in the form of decision trees [6]. For instance, to evaluate inverse reinforcement learning
107 (IRL) methods, users can chose among various domains $\Omega$ and environment dynamics $\mathcal{E}$ and define
108 different ground-truth reward functions $R_\theta$ with parameters $\theta$. Then, users can run Q-learning [42]
109 to obtain the optimal policy $Q_{\pi_\theta}^{\omega}$ for reward $R_\theta$, and add it to Medkit, which can then be used to
110 simulate a batch dataset with demonstrations $\mathcal{D}_{syn,\mathcal{E}}^{\omega}$ for training their IRL algorithm. The recovered
111 policy parameteriaation $\hat{\theta}$ can then be evaluated against the ground truth $\theta$.

Table 1: Summary of related benchmarks key features. Are they focused on the **Medical** setting? Are they designed for **Offline** algorithms? Do they allow **Custom** policies? Do they test how **Robust** algorithms are? Do they incorporate **Non-Markovian** environment dynamics?

| | Benchmark | Medical | Offline | Robust | Non-Markovian | Simulates | Simulated policy |
|---|---|---|---|---|---|---|---|
| RL envs | OpenAI gym [9] | ✗ | ✗ | ✓ | ✗ | Environment Only | N/A |
| | ALE [5] | ✗ | ✗ | ✓ | ✗ | Environment Only | N/A |
| RL and IL benchmarks | RL Unplugged [26] | ✗ | ✓ | ✗ | ✓ | Env. & Policy (Entangled) | Fixed |
| | RL Bench [30] | ✗ | ✓ | ✗ | ✗ | Env. & Policy (Entangled) | Fixed |
| | Simitate [41] | ✗ | ✗ | ✗ | ✗ | Env. & Policy (Entangled) | Fixed |
| | MAGICAL [50] | ✗ | ✗ | ✓ | ✗ | Env. & Policy (Entangled) | Fixed |
| Synth. gen. | TimeGAN [53] | ✓ | ✓ | ✗ | ✓ | Env. & Policy (Entangled) | Fixed |
| | Fourier Flows [2] | ✓ | ✓ | ✗ | ✓ | Env. & Policy (Entangled) | Fixed |
| | Medkit (**Ours**) | ✓ | ✓ | ✓ | ✓ | Env. & Policy (***Disentangled***) | Customizable |

While, as above, users can specify their own policy to roll-out in the environments, we also provide as part of Medkit different types of parameterised policies learnt from the clinicians' policies in the real dataset $D_{real}^{\Omega}$. These built-in policies allow users to easily obtain batch datasets for simulating decision making behaviour with various (customisable) degrees of Markovianity, rationality, counfounding, individual consistency and variation in practice. Details can be found in Section 4.2.

# 3 Alternative Benchmarks and Simulation

Medkit generates *synthetic batch medical* datasets for *benchmarking* algorithms for modelling decision making. There is currently a relative lack of standardised benchmarks for medical sequential decision making and most of the few medical simulators used for evaluation are mathematically formulated as dynamical systems defined by a small set of differential equations (e.g cancer simulator in Gottesman et al. [24], HIV simulator in Du et al. [16]) or are hand-designed MDPs (e.g sepsis simulator in Oberst and Sontag [43], Futoma et al. [22]). Medkit, on the other hand, provides an entire benchmarking suite and enables users to generate data from various medical domains, with realistic environment dynamics and with customisable policy parameterisations. Below, we discuss key differences then with related work, which are summarised in Table 1.

Most benchmarking work has been done outside of the medical domain, in the perhaps most similar work to us [50] present a suite specifically designed to test robustness of imitation learning (IL) algorithms to distributional shifts. Nevertheless, the properties they consider are specifically designed for general robotics tasks than for modelling clinical decision making in healthcare.

Recently offline RL has come more into view and along with it a few benchmarking datasets [26, 30]. These collect state, action, reward tuples of agents deployed in various environments, and despite the focus on RL with the aim to make use of the reward information for some off-policy method like Q-learning, these datasets can be easily used for simple imitation as well. However, at their core they are large collections of recorded trajectories obtained by running trained agents through the live environment. Thus, unlike in Medkit, the end user is not able to specify properties of the policy that are unique to describing human decision-making behaviours such as bounded rationality individual consistency and variation in practice. Indeed this is an issue with any imitation learning benchmark with its origins in RL: due to the reward there's usually only one policy considered the "optimal" one and methods for these benchmarks are mainly evaluated on their ability to achieve a high cumulative reward. This neglects the area of decision modelling [10, 32, 28], where we might be more interested in inference over potentially sub-optimal policies to gain understanding of the human decision-making behaviour. To address this, Medkit enables users to obtain batch medical datasets for various different parameterisation $\theta$ (temperature, markovianity, consistency, bounded rationality, reward) of the policy and the aim is to evaluate algorithms based on how well they can recover $\theta$. Moreover, RL benchmarks focus mainly on Markovian environment dynamics, while Medkit considers the whole history of a patient.

**Generative models for decision making.** Generative models are a long established pillar of modern machine learning [34, 23], though notably they tend to focus on image and text based applications with less focus given to the static tabular data $p(x_s)$ and even less for time-series tabular data $p(\{x_t\}_{t=1}^T)$. Medkit presents as a generative model for the whole process $p(x_s, \{x_t\}_{t=1}^T, \{y_t\}_{t=1}^T)$,

based on the factorisation of equation 1. Importantly this allows for control over the policy, which is very important for the purposes we have in mind, and which traditional methods for synthetic data generation cannot handle normally. Typically to apply generative models designed for static data, for example through normalising flows [15], to this setting it would involve merging all the static features, series features, and actions into one large feature vector. This works especially badly for variable length time series requiring padding and that any relationships between variables cannot be customised. Methods that are specifically designed to work on time series data have been proposed based on convolutions [44], deep Markov models [38] and GANs [53] among others. Generally they model an auto-regressive process - a notable exception being [2] who use a Fourier transform to model time series within the frequency domain, making it inapplicable for sequential generation. Once again though all of these models do not take into account actions (and rarely static features) meaning they have to be absorbed into the series features and cannot be customised.

# 4 Medkit Customisable Scenarios

We describe here the the various domains, policies and environment dynamics we provide in the Medkit package. These can be combined arbitrarily to obtain a large number of different scenarios for batch data generation. Medkit can also live simulate the environment but without reward information is inappropriate for reinforcement learning.

## 4.1 Domains

While Medkit generates *synthetic* data, the machine learning methods used in the generation process are trained on *real* data. This is needed to capture the complexity of real medical datasets and maximise the realism of the scenarios and generated synthetic data. Thus, unlike in the toy medical simulators seen in the literature [43, 16, 24], the batch datasets that can be simulated from Medkit are high dimensional and governed by complex non-linear dynamics, providing a much more realistic environment to test policies in while still maintaining ground-truth information that can be used to evaluate any learnt policy.

Out-of-the-box Medkit contains two medical domains $\Omega$ for which data can be generated, capturing different medical settings: (1) Wards: general hospital ward management at the Ronald Reagan UCLA Medical Center [4] and (2) ICU: treatment of critically ill patients in various intensive care units [33, 19]. While for each domain, the data has undergone pre-processing to de-identify and prevent re-identification of individual patients, we add an extra layer of protection in the form of *differential privacy* [17] guarantees by employing differentially private optimisation techniques when training models, which is readily supported by PyTorch's Opacus library [21]. By ensuring that the generated data is synthetic, Medkit enables wider public access without the risk of sensitive information being inappropriately distributed. Specific details on the state and action spaces for each domain can be found in the Appendix along with details of the real data upon which they are based.

## 4.2 Policies

The key advantage of Medkit is that we separate the environment dynamics from the policy dynamics. This enables us to roll-out customised policies within the environment, and obtain batch datasets where the ground-truth policy parameterisation is know. While users can define their own policy parametrisations, we provide several built-in policies modelling the distribution:

$$p(\vec{y}_T|\vec{x}_T) = \prod_{t=1}^{T} Q_\pi^\Omega(y_t|\vec{x}_t, \vec{y}_{t-1}) \tag{2}$$

By default we might be interested in a policy that seemingly mimics the seen policy in the data as well as possible and so we include powerful neural-network based learnt policies. Of course, as we hope to have conveyed already, the interesting part comes in how the policy seen in the data can be customised in specific ways that are interesting for imitation learning algorithms to try and uncover. As such all policies are constructed in a specific way:

$$Q_\pi^\Omega(y_t|\vec{x}_t, \vec{y}_{t-1}) = \sum_i w_i \frac{e^{\beta_i q_i(y_t|g_i(\vec{x}_t\langle\mathcal{X}'\rangle_i, \vec{y}_{t-1}))}}{\sum_{y\in\mathcal{Y}} e^{\beta_i q_i(y|g_i(\vec{x}_t\langle\mathcal{X}'\rangle_i, \vec{y}_{t-1}))}}$$

that introduces a number of components and properties that Medkit allows us to model and can be controlled simply through the API, the details of which are highlighted below:

1. **Ground-truth Structure** - the policy of a clinician will likely be difficult if not impossible to describe. Even if they could articulate the policy, the information will not be available in the data. Alternatively, we might expect there to be some structure, since for example medical guidelines are often given in the forms of decision trees [12, 49]. An algorithm that uncovers such structure on regular medical data cannot be validated, since we do not know if that inherent structure is in the data or just something the algorithm has picked out - Medkit allows us to provide this ground truth with which we can compare against.

2. **Markovianity** - the common assumption in sequential decision making is usually that the problem can be modelled as a Markov decision process such that for a policy that can be expressed $q(y_t|g(\vec{x}_t, \vec{y}_{t-1}))$ this is constrained so that $g(x_t) = g(\vec{x}_t, \vec{y}_{t-1})$, assuming that the previous observations contains all of the relevant information. With Medkit we can simply model more complicated policies that take into account information much further into the past. We define the Markoviantity of the policy as the minimum time lag into the past such that the policy is equivalent to when considering the whole history: $\inf\{i \in \mathbb{N} : g(\vec{x}_{t-i:t}, \vec{y}_{t-1-i:t-1}) = g(\vec{x}_t, \vec{y}_{t-1})\}$.

3. **Bounded Rationality** - clinicians may not always act optimally based on all the information available to them. In particular they may overlook some specific variables as though they are not important [36]. We can model this in Medkit by masking variables going into the policy model so that $q(y_t|g(\vec{x}_t, \vec{y}_{t-1})) = q(y_t|g(\vec{x}_t\langle\mathcal{X}'\rangle, \vec{y}_{t-1}))$, where $\mathcal{X}'$ is a subspace of $\mathcal{X}$ and $\vec{x}_T\langle\mathcal{X}'\rangle = x_s \cup \{\text{proj}_{\mathcal{X}'} x_t\}_{t=1}^T$. Here, the dimensionality of $\mathcal{X}'$ relative to $\mathcal{X}$ given as $\dim \mathcal{X}' / \dim \mathcal{X}$ can be used as a measure of the agent's rationality.

4. **Individual Consistency** - some clinicians are very consistent, they will always take the same action given a specific patient history. Others are more stochastic, they'll tend to favour the same actions but might occasionally choose a different strategy given a "gut feeling" [18]. Medkit can model this with the temperature of the Boltzmann distribution given in the output of all of the policies. Formally, for policies of the form $p(y_t|\vec{x}_t, \vec{y}_{t-1}) = \exp \beta q(y_t|g(\cdot))/\sum_{y \in \mathcal{Y}} \exp \beta q(y|g(\cdot))$, the inverse temperature $\beta \in \mathbb{R}_+$ measures the individualised variability of an agent, where $\beta = 0$ means that the agent acts completely at random while $\beta \to \infty$ means that the agent is perfectly consistent (i.e. their actions are deterministic).

5. **Variation in Practice** - often (essentially always) medical datasets are not the recordings of a single clinician's actions but of a mixture or team that consult on an individual patient [51]. With Medkit we can model this effectively using the `Mixture` policy, which takes any number of policies and a mixing proportion to generate a new mixture policy. Formally, a mixture policy is given by $p(y_t|\vec{x}_t, \vec{y}_{t-1}) = \sum_i w_i q_i(y_t|g(\vec{x}_t, \vec{y}_{t-1}))$ where $\{w_i\}$ are the mixing proportions such that $\forall i, w_i > 0$ and $\sum_i w_i = 1$, and $\{q_i(\cdot)\}$ are arbitrary base policies.

These different policy parameterizations that are in-built into Medkit are specific to scenarios that commonly arise in medicine [18, 51, 36], which is the domain application we consider in this paper. However, note that the main contribution of Medkit is to provide a framework for obtaining customizable policies. Thus, users could also incorporate different types of policies if needed.

## 4.3 Environments

The environment dynamics capture how the patient's covariates evolve over time given their history, interventions and the patient's static features. From the proposed factorisation in Equation (1), to estimate the environment dynamics, we model the following conditional distribution in two parts:

$$p(\vec{x}_T|\vec{y}_{T-1}) = \underbrace{P_{\mathcal{E}}^{\Omega}(x_s, x_1)}_{\text{Initialisation}} \prod_{t=2}^{T} \underbrace{P_{\mathcal{E}}^{\Omega}(x_t|f_{\mathcal{E}}(\vec{x}_{t-1}, \vec{y}_{t-1}))}_{\text{Auto-regression}}, \tag{3}$$

allowing for sequential generation of patient trajectories. For all environments, we model $P_{\mathcal{E}}^{\Omega}(x_s, x_1)$ using a Variational Autoencoder [34],as a powerful generative model that can handle a mixture of continuous and discrete variables. For the auto-regressive part, to capture a diverse set of the realistic dynamics of medical datasets, Medkit contains environments that are (1) directly modelling the patient history (T-Force and CRN) and (2) building latent variable models (CSS and SVAE). We describe the models in this section but full details (e.g. on learning) are given in the Appendix.

**Directly modelling the patient history.** This relates to attempting to model:

$$p(x_t|\vec{x}_{t-1}, \vec{y}_{t-1}) = P_{\mathcal{E}}^{\Omega}(x_t|f_{\mathcal{E}}(\vec{x}_{t-1}, \vec{y}_{t-1})) \tag{4}$$

directly, or more specifically that $p(x_t|\vec{x}_{t-1}, \vec{y}_{t-1})$ is some $\Theta$ parameterised distribution where $\Theta = f(\vec{x}_{t-1}, \vec{y}_{t-1})$ is a function of the history only. For the simplest environment model, we use a recurrent neural network trained with teacher forcing [52] (**T-Force**) to directly approximate this function. The network is made up of LSTM units [27] followed by fully connected layers with ELU activations [13] and is trained to maximise the likelihood of the next observation given previous observations and interventions. This defines a factorised Gaussian and Bernoulli distribution over the continuous and binary covariates respectively with the parameters predicted by the network.

Additionally we extend this method by replacing the LSTM network with the Counterfactual Recurrent Network (**CRN**) of Bica et al. [7]. CRN is a causal inference method that learns balancing representation of the patients' histories to remove the time-dependent confounding bias present in observational datasets. This allows the network to more principally be used for making counterfactual predictions which is what our model for the environment dynamics needs to do when estimating the next state of a patient under different possible interventions specified by the policy $Q_{\pi}^{\Omega}$.

**Building latent variable models.** We also build environment dynamics where the observations are driven by a *hidden* true state of the patient. Formally, we assume the features $\vec{x}_T$ are driven by some evolving latent state $\vec{z}_T = \{z_t\}_{t=1}^{T}$, $z_t \in \mathcal{Z}$ that is not seen in the data by modelling a factorisation given by:

$$P_{\mathcal{E}}^{\Omega}(x_t, z_t|f_{\mathcal{E}}(\vec{x}_{t-1}, \vec{y}_{t-1}, \vec{z}_{t-1})) = \underbrace{P_{\mathcal{E}}^{\Omega}(x_t|z_t, x_s)}_{\text{Emission}} \times \underbrace{P_{\mathcal{E}}^{\Omega}(z_t|f_{\mathcal{E}}(\vec{x}_{t-1}, \vec{y}_{t-1}, \vec{z}_{t-1}))}_{\text{Transition}}. \tag{5}$$

We include as part of Medkit two additional environment dynamics models for the separate cases when $|\mathcal{Z}|$ is finite or uncountable, as both can usefully represent patients in the medical context. For $|\mathcal{Z}|$ finite the latent $z_t$ variables then might represent distinct progression "stages" or various classifications of a disease. Discrete separation like this is well established in both clinical guidelines and models for a range of cases including transplantation in patients with CF [8], the diagnosis of Alzheimer's disease [45], and cancer screening [47]. Accordingly we use the Attentive State-Space model of [3] to build an attention-based, customised state-space (**CSS**) representation of disease progression. This environment model accounts for static features and allows Medkit users to customise the attention mechanism. Given a discrete latent space, the transitions are parameterised with baseline transition matrices for each action averaged over attention weights on previous timesteps. The emission distribution allows for a flexible representation: let $p_{\psi}(x_t)$ be any distribution with support over $\mathcal{X}$ and parameter(s) $\psi$ (for example some Gaussian mixture) then we let:

$$p(x_t|z_t, x_s) = p_{\psi^*}(x_t), \quad \text{with } \psi^* = f_{\gamma}(z_t, x_s). \tag{6}$$

We take $f_{\gamma}$ to be a $\gamma$-parameterised function approximator to output the parameters of the emission distribution given the current state and static features of the patient - a standard choice being an MLP that takes in the concatenation of $z_t$ and $x_s$. This alleviates a common problem with state-space models where the observations are ultimately drawn from some finite mixture of distributions of order $|\mathcal{Z}|$, as now the dependence on the static features allows for a very flexible output. The CSS dynamics model allows Medkit users to post-hoc customise the number of states and the Markovianity of the environment through the attention mechanism (e.g users can pass a vector that specifies exact weights or an integer representing the number of states back to look.)

While a discrete representation of hidden states is convenient for interpretation, it does simplify the problem. It is unlikely that all of the relevant features of a disease can be adequately captured by a discrete characterisation - it would seem that in reality diseases evolve gradually and without step-change. Therefore, to further improve the realism of the generated trajectories, we also include as part of Medkit's environments a deep continuous state space model that extends VAEs in a sequential manner (**SVAE**). Principally now we consider a continuous latent state with $\mathcal{Z} = \mathbb{R}^d$. This then allows for more flexibility in the transition dynamics, in particular by making use of neural architectures. An encoder network predicts the approximate posterior over the latent variables and we employ essentially the same method as for teacher forcing in order to model dynamics in the latent space. With a joint optimisation scheme, we learn a representation that generates the observations well but also captures the features relevant for the transitions. This expressiveness allows for a higher fidelity model than the custom state-space but however comes at the cost of interpretable structure which we have established may be useful should algorithms be designed to uncover such things.

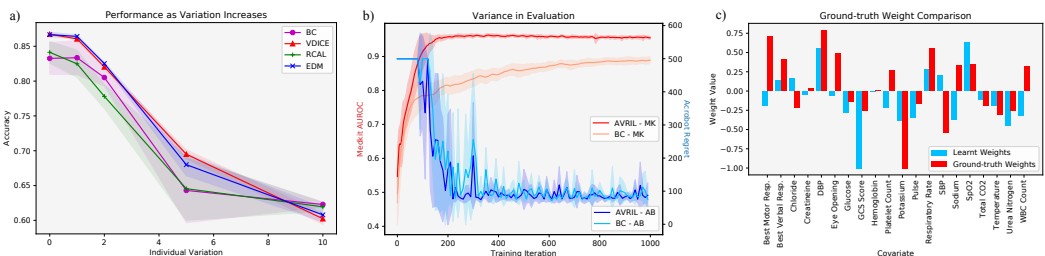

Figure 3: **Exploring Medkit Practically**. Example benefits of Medkit for exploring and benchmarking imitation learning algorithms.

**Modelling hidden confounding.** A common assumption, that is likely not true in practice, is that there are no hidden confounding variables in the environment. Medkit allows us to introduce and control these by using a full set of variables to generate both the actions and the observations but restrict the visibility of some such that they become hidden. While the overall generative process $p(\vec{x}_T, \vec{y}_T)$ is left unchanged, only a partially-hidden dataset $\mathcal{D} = \{\vec{x}_T\langle\bar{\mathcal{X}}'\rangle, \vec{y}_T\}$ is provided to the user, where $\mathcal{X}'$ is a subspace of $\mathcal{X}$ and $\vec{x}_T\langle\mathcal{X}'\rangle = x_s \cup \{\text{proj}_{\mathcal{X}'}\, x_t\}_{t=1}^T$. Here, the dimensionality of $\mathcal{X}'$ relative to $\mathcal{X}$ given as $\dim \mathcal{X}'/\dim \mathcal{X}$ can be used as a measure of the overall confoundedness.

## 5 Practical Demonstrations

In this section we explore some examples of the benefits of using Medkit compared to existing benchmarks as well as highlight some potential use cases, in particular how Medkit allows for consistent and systematic evaluation along with useful ground truth information.

**Different reactions to shifting policies.** The current literature on imitation learning focuses on very different environments to those found in the medical setting and consequently algorithms may not be evaluated against, or designed to be appropriate for, the quirks of medical data. For example in Figure 3a we plot the performance of algorithms as the consistency of the policy varies, in particular we use: Behavioural Cloning (**BC**) with a deep Q-network; Reward-regularized Classification for Apprenticeship Learning (**RCAL**) [48], where the network is regularised such that the implicit rewards are sparse; ValueDICE (**VDICE**) [37], an offline adaptation of the adversarial imitation learning framework; and Energy-based Distribution Matching (**EDM**) [31] that uses the implicit energy-based model to partially correct for the off-policy nature of BC. What is interesting is not that performance degrades - this is of course to be expected, but rather that the comparative ranking of algorithms changes as a function of the consistency. In particular BC performs the worst (although there is little between them) in the ends up outperform the rest on average when the variation is highest, suggesting some of the more complicated algorithms are not robust to these kinds of policies.

**Enabling consistent evaluation.** Common RL benchmarks like Atari experience very large variances in the accumulated reward an agent obtains when deployed in the environment, especially when the reward is sparse. This can make evaluation and ranking of agents tricky or at least require a large number of runs in the environment before the variance of the estimator suggests the results are significant. In Figure 3b we demonstrate this problem in an even simpler context comparing BC to the AVRIL algorithm of [11], a method for approximate Bayesian IRL, in the simple Acrobot environment where the aim is to swing up a pendulum to a correct height. On the right y-axis we plot the accumulated regret over training of the two agents, and large inconsistencies in return can be seen so that it is not clear which of the agents is better. Comparatively on the left y-axis we plot the AUROC on a held out test set as we train on Medkit data, here evaluation is much more consistent and statistically significant, demonstrating clearly which algorithm is performing better.

**Ground-truth knowledge comparison.** While in the end it only really matters how an algorithm performs when deployed in the real world, it is challenging to only use real data to validate them. This is since you run into the key problem that you will not have any knowledge of the ground truth behind decisions and so methods that claim to gain insight into such areas cannot possibly be evaluated appropriately. On the other hand simulating data in Medkit allows us to do exactly this, and we can compare inferences from an algorithm to underlying truth in the generating process. A toy example is shown in Figure 3c where we compare the weights of a linear classifier trained on Medkit data to those of the true underlying policy, representing the relative feature importances for the policies.

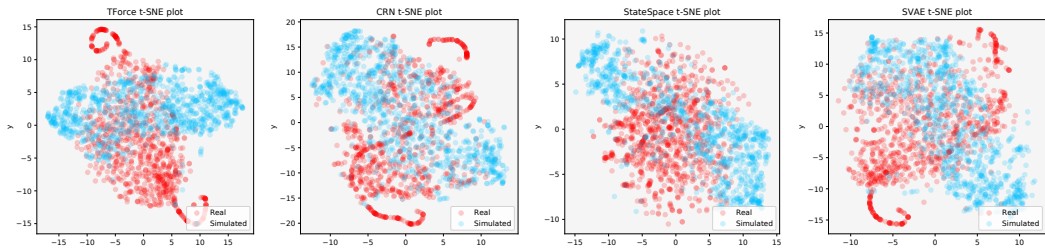

Figure 4: **t-SNE plots** For each policy in the Ward environment we generate simulated data. We then apply t-SNE and project the real and simulated data into two components, which is plotted.

**Validating realism.** It is also of interest to quickly check that we are not generating completely unrealistic trajectories, rather ones that capture appropriate properties that will be useful for users. We thus provide comparisons of the available environment models in Medkit. In particular for each combination we show in Table 2: the *Predictive Score*, a classical "train on synthetic - test on real" evaluation where a network is trained on the synthetic dataset and applied to a held out test set of the real data, where the performance is reported; and the *Discriminitive Score*, where a classifier is trained to distinguish between the real and synthetic data, and the AUROC of this task on a held out test set is reported. In aid of visualisation we also provide in Figure 4 a set of t-SNE plots [40] overlaying the real and synthetic data. These metrics are standard in the synthetic data literature [53] and reflect the usefulness of the synthetic data as a *replacement* for real data.

Please note though that the highest possible fidelity is not the point of Medkit: unlike traditional synthetic data, the datasets we produce are *not* meant to be used as a substitute for real data in training machine learning algorithms. Rather we would like to produce *realistic* data that reflects the difficulties of the medical setting and can be used for development and benchmarking of algorithms. Additionally, by introducing customi-

Table 2: **Predictive and Discriminative Scores.** Scores reported on the different environments for the Wards domain.

|  | $|\mathcal{Y}|$ | **T-Force** | **CRN** | **CSS** | **S-VAE** |
|---|---|---|---|---|---|
| Pred.↑ | 2 | $0.67 \pm 0.05$ | $0.94 \pm 0.01$ | $0.94 \pm 0.01$ | $0.93 \pm 0.01$ |
|  | 4 | $0.62 \pm 0.02$ | $0.85 \pm 0.01$ | $0.86 \pm 0.01$ | $0.86 \pm 0.02$ |
|  | 8 | $0.61 \pm 0.05$ | $0.85 \pm 0.03$ | $0.89 \pm 0.02$ | $0.87 \pm 0.04$ |
| Disc.→ | 2 | $0.41 \pm 0.03$ | $0.23 \pm 0.02$ | $0.19 \pm 0.03$ | $0.22 \pm 0.04$ |
|  | 4 | $0.41 \pm 0.05$ | $0.24 \pm 0.04$ | $0.19 \pm 0.04$ | $0.23 \pm 0.04$ |
|  | 8 | $0.37 \pm 0.07$ | $0.22 \pm 0.03$ | $0.20 \pm 0.03$ | $0.20 \pm 0.02$ |

sations into the generative process, we will naturally see departures from real data, but given our goals this is not a problem. Nevertheless, the high predictive scores show that Medkit is successfully capturing important trends in the real data that are useful for prediction, while the discriminative scores and t-SNE plots confirm that we are not producing trajectories that are unrepresentative.

## 6 Discussion

**Limitations and Societal Impact.** As a synthetic data generator, Medkit is inherently limited by the power of the individual models used and their ability to accurately model outcomes given specified policies. This is not such a problem when the focus is on inference over the policy though, as is the focus in decision modelling. Additionally, Medkit is easily extendable when new, more powerful, models become available. With Medkit our aim is to provide a platform allowing for better development of decision modelling algorithms, the societal impact thus very much depends on the potential use of such algorithms, for example, they could be used to misrepresent an individual's position or identify biases that could be exploited. By focusing on clinical decision support, we hope to promote a much more beneficial approach.

**Conclusions.** We have presented the Medkit-Learn(ing) Environment, a benchmarking suite for medical sequential decision making. As with many software libraries, the work is never done and there are always new features that can be added. Indeed we can, and intend to, always continue to add more tools and algorithms to be beneficial for the community. One important future area that Medkit could make an impact in is causality - an area where more than ever synthetic data is important such that we can actually evaluate the counterfactuals that are inherently missing from real data, and much can be done to simulate data for individualised treatment estimation for example. Overall though our aim with Medkit is to advance the development of algorithms for *understanding*, not just imitating, decision making so that we can better support those high-stakes decisions such as in the clinical setting without replacing the crucial human aspect needed when the problem is so important.

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
