# OpenReview forum: "The Medkit-Learn(ing) Environment: Medical Decision Modelling through Simulation"
_NeurIPS.cc/2021/Track/Datasets_and_Benchmarks/Round1 — Submitted to NeurIPS 2021 Datasets and Benchmarks Track (Round 1)_

### Official Review · Reviewer_ULYY · 2021-07-03
**This work contributes a benchmark environment for clinical sequential decision making to capture real-world challenges of training models for clinical decision support. The problem is well motivated and the authors have come up with a reasonable solution to create such a challenging benchmark. My main concern is with the modeling choices made for modeling the environments, which I believe could be significantly improved.**

**Rating:** 5
**Confidence:** 4

**Strengths:**

1. ML-based policies and interventions learned on clinical data are fairly challenging to evaluate on the data itself and real-world experimentation is often not possible/unethical. Such a benchmark (which is fairly general) can prove very valuable to the community interested in building models for clinical decision support for proof-of-concept demonstrations of their support models.

2. The methodology of designing the environment is clear and looks reasonable to me, particularly the non-markovian policies and auto-regression (markovian environments). I do suggest authors make that distinction in Table 1 unless I missed something (i.e. only policies can be non-markovian, environment dynamics cannot).

3. The main utility of the benchmark seems to be in its ability to evaluate RL algorithms and account for realistic clinician behavior in policies (like practice variation etc).



**Weaknesses:**

My main concern is the way environments are modeled. I understand the need for realistic data. Authors have chosen to model environments with methods like recurrent neural networks, "counterfactual" recurrent neural networks and latent variable models like attention-based state-space models and variational auto-encorders.

Designing the environments in this way clearly biases the benchmark and ability to evaluate them to certain methods over others. I have multiple concerns about this that I have listed below:

i) At a conceptual level, it is probably better to describe variability of environments in terms of statistical variations rather than specific model types used. By that I mean it is more meaningful to users of such a benchmark to be able to claim that a particular policy/method works well on, say i) non-stationarity (and what kind of non-stationarity), ii) heteroskedasticity (and what kind), iii) multi-modality, iv) heterogeneity, instead of saying something general about a neural network based environment.

ii) Model-based methods in RL cannot be realistically and properly evaluated with such environments, all one has to do is create parametric models of the dynamics that match the environment property and they will mis-specify the model far less often compared to those that don't. If the benchmark is not designed for model-based evaluation, authors should clearly say so in the introduction. This can clearly bias which methods do better than others. I strongly suggest authors provide complex parametrized models that add noise over real data to create semi-synthetic data-generators for the environment instead of deep network based environments. It is not even clear whether such deep networks are good proxies for realistic clinicial environments. But statistical notions of choices of environments can certainly help understand limitations of the environment without biasing methods. It will make the benchmark more usable in my opinion.

iii) Relying on semi-synthetic data + specific parametrization of the *distributions* rather than deep learning models of the environment will also significantly avoid bias in the generate samples. If a certain population or disease type is more prevalent in the data, it clearly biases the environment in a manner that's not appealing for strong evaluation.



**Additional Feedback:**

If authors can clarify the concerns I have raised above, particularly in weaknesses, in detail, I am happy to reconsider the score.

**Clarity:**

The paper is fairly well written and motivated. Please just check for minor typos.

**Correctness:**

i) Please see my concerns above on weaknesses. I believe this is a very strong limitation of the current benchmark and something to address to improve the utility of the benchmark.

ii) Why exactly will $q(y|.)$ in Equation for $Q_{\pi}$ - \begin{equation}
Q_{\pi}^{\Omega}\left(y_{t} \mid \vec{x}_{t}, \vec{y}_{t-1}\right)=\sum_{i} w_{i} \frac{e^{\beta_{i} q_{i}\left(y_{t} \mid g_{i}\left(\vec{x}_{t}\left\langle\mathcal{X}^{\prime}\right\rangle_{i}, \vec{y}_{t-1}\right)\right)}}{\sum_{y \in \mathcal{Y}} e^{\beta_{i} q_{i}\left(y \mid g_{i}\left(\vec{x}_{t}\left\langle\mathcal{X}^{\prime}\right\rangle_{i}, \vec{y}_{t-1}\right)\right)}}
\end{equation}
model ground-truth structure that can be customized and verified? Can the authors elaborate a bit? I understand it helps generating data, but doesn't help me understand why it captures structure.

**Documentation:**

Yes intended use, code is available, with a basic documentation/readme. I did not see a maintenance plan. Sufficient detail is provided for reproducibility

**Ethics:**

It wasn't clear to me how the real data was utilitized for the domains, and its access for other users. Clarification in this regard should be sufficient for ethics verification in my opinion.

**Relation To Prior Work:**

Related work, existing benchmarks are well contextualized.

**Summary And Contributions:**

This work proposes a benchmark for clinical sequential decision making. The simulated environment of course consists of the underlying dynamics model and policy model, both of which can be independently modeled. The benchmark provides flexibility in terms of policy classes that can be used to sample synthetic data from two types of domains (ICU and Ward). The environment model is captured using various deep learning methods trained on real-world data that can then be used to sample synthetic data under custom policies. Benchmark is evaluated on imitation learning baselines and discriminatory performance comparison of synthetic and real data samples. Policy functions can be customized to modify clinician/behavior policy, markovianity of the policy, practice variability and other factors. Environment is modeled with different choices like LSTMs, recurrent counterfactual networks, latent variable models.

---

> ### Author Response · Authors · 2021-07-12
> **Author response**
>
> Thank you for the review and your thoughtful comments, we appreciate your suggestions for improving the manuscript. Please see below some specific responses to points raised in your review.
>
> First, we’d just like to emphasise that it is the customisation of the policies that we are most focused on. In order to evaluate decision modelling algorithms, we need to be able to generate and compare against a range of customisable policies in order to do effective benchmarking and ablation studies. The environment dynamics models are crucial as they allow this to happen by learning the important conditional distributions to sample states from. However, testing algorithms on their ability to infer properties about the environment transitions is less important to us than the focus on the policies - which is why at the moment the majority of customisation is focused on the policies. However, this is still extendable and we intend to increase all of the customisation options in the future.
>
> “At a conceptual level, it is…”
> We agree that it is definitely better to describe the environments as such - and this is definitely the approach we have taken with the policies (being able to control Markovianity, consistency, variation, etc.).
> The environment models were also chosen to act in a similar way, as all these models simply define a transition kernel given a history and selected action. T-Force and CRN both represent general fully-observed transitions, i.e. so that the transition is just some function of the visible history - while the CSS and SVAE contain a latent state that makes the transitions only partially explainable from the visible history, making an interesting distinction that should favour different algorithms. The fundamental difference between discrete and continuous latent states also offers useful customisation for users to test algorithms against.
> We will also introduce more post-hoc customisation options for the environment models (like there are currently for the policy models). The CSS can be customised by merging states or altering the attention mechanism as described in section 4.3, but we can easily add additional customisations to all of the different types of models, as you suggest, by injecting customised noise into the transitions and introducing multimodality with multiple mixture models. We plan to push some of these changes to the codebase very soon.
>
> “Model-based methods in RL…”
> It is worth saying that model-based RL is certainly not the focus of this work, as we’re much more interested in decision modelling based on the policy of a demonstrator - we shall aim to make this much clearer, as you suggest, in the introduction.
> The problem we see with implementing complex parameterised models that add noise on top of real data is that firstly any parameterisation will likely be easier for an RL method to learn than a neural network which is essentially a non-parametric function approximator that maps to the parameters of a conditional distribution. Secondly, it would be hard to use the real data appropriately since we have to model the conditional distributions given a history, and in almost every case during simulation, there will be no samples in the real data from that specific conditional. This is why we use more flexible function approximators to learn the parameters of these conditional distributions. Of course, if we have misunderstood your suggestion please let us know and we can discuss it further.
>
> “Relying on semi-synthetic data…”
> We agree that modelling the distributions will reduce bias, which is our aim in the models we picked. We have inspected the generative models to try to ensure they are not producing severely biased data and we would also emphasize that the data generated by Medkit should not be used for scientific discovery. For example, it should not be used to say anything about how real doctors treat patients in the ICU nor how real patients will react to treatment. It is rather intended to be used with a focus on inspecting algorithms for decision modelling.
>
> “Why exactly will q(.) in Equation…”
> This was simply to demonstrate the factorisation that allows for all of the other customisations - it’s definitely not clear from the overall equation why this models ground truth structure. What we are showing here is that q(.) itself can be any function (i.e. a decision tree, linear model, set of symbolic rules etc.) but that this is embedded in a bigger structure.
>
> “It wasn't clear to me how the real data…”
> For the domains, the real data was only used to define the number and names of the variables essentially. Most of its use comes in the pre-trained environment and policy models. No real data is, or will be, made available to end-users, they only have access to trained models.

---

> > ### Comment · Reviewer_ULYY · 2021-07-21
> > **Thank you for your response**
> >
> > I thank the authors for their response to my concerns. I get that the main goal is about customized policies and evaluating decision modeling under custom data collected under different behavior policies, which yes in real data is unavailable. I do think it is worth commenting on and justifying the choices made for transition dynamics a bit more. In essence it is still unclear to me why I can't game this benchmark by simply modeling the dynamics precisely the same way as you have. I guess that is my main disconnect, and I would really appreciate a discussion in the draft on this. I am assuming authors will do this in addition to highlighting that model-based RL policies is not the focus of this evaluation. Unless I am misunderstanding the usage, these nuances are important for the benchmark to be truly reliable for even inspecting policies? I have increased my score based on this discussion.

---

### Official Review · Reviewer_73jP · 2021-07-04
**Interesting data generator for medical decision making with unclear practical use.**

**Rating:** 7
**Confidence:** 3

**Strengths:**

- Very well written manuscript, including the definition of a scenario, domain, environment and policy and the graphical model.

- The authors have done an excellent job of framing their new package in the context of other comparable ones, and highlighting key different contributions.  The authors have made it clear why their work is significant.

- The authors have created highly customisable scenarios, with detailed descriptions on the different domains, policies (along with the different API aspects), environments (patient history, latent variable models). This increases the relevance to the broader community as well as well accessibility.

- The authors have considered ethical and social implications, and have presented a clear focus for their work without overstating any claims.

**Weaknesses:**

- Brief practical demonstrations. I suggest the authors include demonstrations using the ICU data, and expand on the practicality of their package. Given that minimal comparable works exist for the medical context, I am concerned that the broader community may not be interested in this very interesting work or find it inaccessible otherwise. Demonstrating more practical examples would be one way to mitigate that.

- A lack of complete examples (in code) may limit user adoption and usage. For example, the authors outline a potential workflow in 2.2, but no similar example exists in their code repository as far as I can tell.

- The validation of realism is superficial and not compelling.

**Additional Feedback:**

- There appear to be multiple contributions here. It's unclear to me which are the most significant and which are incremental in nature. Can the authors elaborate on the relative significance of being a) non-Markovian, b) having a distentangled environment and policy, and c) having a customizable simulated policy? (Relative to one another)

- What would the process be for outside contributors wanting to add new medical domains?

**Clarity:**

- Consistency on the capitlization of Markovianity (pg 4, line 144)
- While I appreciate the colour coding in 4.2, some colours are not easily distinguished for those of us that are colour blind. I suggest another form of formatting or different colours.
- Figures 3 and 4 is relatively hard to read at normal zoom.

Otherwise, the paper is very well written given the topic matter and the many concepts being introduced.

**Correctness:**

The package presented here is logically sound and correct. The way the authors have presented this work is sound.

However, there appear to be missing definitions for functions f and g. The first mention of these functions appears on page 3, section 2.1, line 77 as far as I can tell. For completeness, please add in these definitions.

**Documentation:**

Documentation is a difficult section to evaluate the authors' work. Their work is not a dataset itself, nor is it pure benchmarking. I judge documentation by thoroughness of their manuscript's description and the code documentation. The manuscript description and detail is strong, including details about differential privacy. The code documentation is weak to moderate, as many classes lack thorough documentation or are not self-describing.

**Ethics:**

The authors do discuss the potential ethics, although it seems they differ any consequences of their work to rely on the models themselves.

I would ask that, if Medkit is a data generator, could it not generate biased or inappropriately skewed data?

**Relation To Prior Work:**

The authors have clearly positioned their package against others, including going beyond medical only applications to look at numerous other works. The authors have then identified key factors to consider for all of those and address the shortcomings in previous contributions.

**Summary And Contributions:**

The authors present Medkit-Learn(ing) Environment, a modular python package for offline sequential decision making. This package enables the creation of simulated medical setting, with high customizability and controllability. The authors outline each of aspect of their contributions and provide some initial demonstrations.

---

> ### Author Response · Authors · 2021-07-12
> **Author Response**
>
> Thank you for the review and your thoughtful comments, we appreciate your suggestions for improving the manuscript. Please see below some specific responses to points raised in your review.
>
> “Brief practical demonstrations…”
> Thank you for the suggestions - we will incorporate some more experiments specifically on the ICU data in the additional space provided and in the appendix. We certainly hope that it is not just the medical community that will find this tool useful as the challenges we highlight do exist elsewhere, it is just that they are particularly prevalent in this area.
>
> “A lack of complete examples…”
> We will add some tutorial notebooks and example scripts to the code repository to further flesh out the examples given there, we agree this should be very helpful to encourage user adoption.
>
> “Missing definitions for functions f and g…”
> Thank you for catching this, we shall make sure to define these properly in the paper, f and g simply represent abstract functions that will be defined according to the specific model used for the policies and environments, but we shall make this clear in the paper.
>
> “Could it not generate biased or inappropriately skewed data?”
> It is definitely possible, as with any generative model, however, we have inspected all of the models to try to ensure that this is not happening. We would emphasize that the data generated by Medkit should not be used for scientific discovery though - for example, it should not be used to say anything about how real doctors treat patients in the ICU nor how real patients will react to treatment. It is rather intended to be used with a focus on inspecting algorithms for decision modelling
>
> “There appear to be multiple contributions…”
> In our opinion, the order is c-b-a. Primarily the goal here is for understanding the decision-maker, which means we really want to focus on (c) the policy being followed and how customisations impact the quality of inference. This is enabled by (b), disentangling the policy and environment dynamics, as this means we can separately model the policy and fully customise it. The non-Markovianity (a) is one aspect that can be customised but we would not consider this aspect to be particularly significant compared to the other customisations possible.
>
> “What would the process be for outside contributors wanting to add new medical domains?”
> This should be relatively easy. The first way would be for them to provide us with a copy of the medical data, we could then add the details and train the models before adding them to the package. If they would rather not share the data, then the second way would be for the outside contributors themselves to fork the repo, define the domain, and train the models using the codebase before making a pull request to us. We would then make sure the pre-trained models work appropriately before merging into the package.
>
> Finally, thank you for pointing out the issues with the colours used - we will adjust the colour palette to be more colour-blind friendly.

---

> > ### Comment · Reviewer_73jP · 2021-07-12
> > **Good response from the authors.**
> >
> > I thank the authors for the specific responses. I have no further comments. I look forward to reading the full camera-ready version.

---

### Official Review · Reviewer_h6s3 · 2021-07-04
**Useful simulator for modelling sequential decision making in different medical environments.**

**Rating:** 7
**Confidence:** 2
**Clarity:** This paper was written quite clearly.

**Strengths:**

This paper is very well-written and provides a new simulator that is much more flexible than many of the ad-hoc simulators that have been created for medical sequential decision making studies in the ML4H community. The data generating process is very clear and the demonstrations provide good insight into potential uses for the simulator.

**Weaknesses:**

The use of differential privacy is mentioned in the construction of the synthetic datasets. This was barely discussed in the paper but seems to play an important part in ensuring that the data is protected. For example what epsilon, delta values were chosen and what is the effect of this on the final synthetic datasets that are generated? Additionally, it would be great to see more empirical comparisons of this simulator to some of the ad-hoc simulators. For example, how trustworthy are results on these synthetic datasets compared to dynamics based simulators used in the sequential decision making ML4H literature?

**Additional Feedback:**

- Spelling mistake of parameterization on Line 111 Page 3


**Correctness:**

All statements appear to be correct to me.


**Documentation:**

This dataset is well documented in the associated Github.


**Ethics:**

I don’t see major ethical concerns except for the lack of insight into the use of differential privacy? But this is something that can be easily rectified with some exposition from the authors. This does not constitute concern for whether the benchmark should be released.


**Relation To Prior Work:**

This paper did a great job of comparing the differences of this simulator to other simulators at a high level in the table provided. It would be great to seem some empirical comparisons.

**Summary And Contributions:**

This study presents “Medkit-Learn(ing) environment” which gives researchers the ability to generate a variety of synthetic datasets to advance sequential decision making (offline RL, IRL, reward design) in medical machine learning. The study describes the data generating process used to provide datasets that are customized with different environment and policy dynamics which is different from other similar simulators which entangle these two dynamics.Finally, some practical tasks and evaluations are demonstrated using the library.

---

> ### Author Response · Authors · 2021-07-12
> **Author response**
>
> Thank you for the review and your thoughtful comments, we appreciate your suggestions for improving the manuscript. Please see below some specific responses to points raised in your review.
>
> “The use of differential privacy…”
> We would like to emphasize that all of the real data used to train models within Medkit had already undergone a full pre-processing procedure from their respective data guardians to de-identify the data and ensure the privacy of individuals. As a consequence, the added differential privacy layer that we add is not essential but does serve as an extra “insurance policy” just in case. We will however clarify in the appendix the chosen parameters as well as an ablation study of how they affect the generated data as in Section 5.
>
> “It would be great to see more empirical comparisons…”
> This is a great point - in the extra space provided with the camera-ready version, we will provide some more empirical comparisons with existing simulators to highlight empirically some of their shortcomings.

---

> > ### Comment · Reviewer_h6s3 · 2021-07-13
> > **Thanks for response!**
> >
> > I thank the authors for their responses and will keep my score as is based on these responses.

---

### Decision · Program_Chairs · 2021-07-27

**Decision:**

Reject

**Comment:**

The reviewers feel that the new “Medkit-Learn(ing) environment” is well motivated and the authors have come up with a reasonable solution. The discussion has converged to a positive evaluation by the reviewers R1 and R2, but R3 still has concerns about the modeling choices made for modeling the environments, which could be significantly improved. In view of that, the authors may need to expand them in the next version.